# Impact of COVID-19 Infection, Vaccination, and Serological Response in Immune Thrombocytopenic Purpura Patients: A Single-Center Global Analysis

**DOI:** 10.3390/biomedicines10112674

**Published:** 2022-10-23

**Authors:** Cristina Dainese, Federica Valeri, Marco Bardetta, Carola Sella, Annamaria Porreca, Alessandra Valpreda, Fabrizia Pittaluga, Giulio Mengozzi, Benedetto Bruno, Alessandra Borchiellini

**Affiliations:** 1Regional Reference Center for Thrombotic and Haemorrhagic Disorders of the Adult, Department of Hematology and Oncology, Hematology Divison, University Hospital City of Science and Health Molinette, 10126 Turin, Italy; 2Department of Medical, Oral and Biotechnologies Sciences, University of Chieti-Pescara, 66100 Chieti, Italy; 3Clinical Biochemistry Laboratory, University Hospital City of Science and Health Molinette, 10126 Turin, Italy; 4Laboratory of Microbiology and Virology, University Hospital City of Science and Health Molinette, 10126 Turin, Italy; 5Hematology Division, Department of Hematology and Oncology, Azienda Ospedaliera Città della Salute e della Scienza di Torino-Molinette, 10126 Turin, Italy

**Keywords:** immune thrombocytopenic purpra, ITP, SARS-CoV-2, COVID-19, vaccine, serological assay

## Abstract

Both SARS-CoV-2 infection and vaccination have raised concern in immune-mediated diseases, including immune thrombocytopenic purpura (ITP) considering risk of de novo ITP development and ITP recurrence. Here, we report on data from a single-center retrospective–prospective collection aiming to evaluate platelet (plt) dynamics in patients (pts) with chronic ITP after COVID-19 infection (before and after vaccination) and after the first, second and third vaccine doses. Furthermore, we analyzed the serological response after the first two doses of COVID-19 vaccination. A total of 64 pts currently followed for chronic ITP who experienced COVD-19 infection and/or vaccination with an available plt count before and after such events were included in the analysis. A low incidence of ITP exacerbation following vaccine sessions (6–16%) was observed in comparison with a high frequency of exacerbation and rescue treatment necessity after COVID-19 infection in unvaccinated pts (83%). Moreover, the lower ITP exacerbation rate observed in infected pts previously vaccinated (18%) suggests further protective effects in this population. Finally, a high seroconversion rate was observed, confirming data reported in previously published studies on immune cytopenia and rheumatological diseases, but more evidence is awaited to establish the clinical impact of serological response.

## 1. Introduction

Immune thrombocytopenic purpura (ITP) is an autoimmune disease that is commonly idiopathic but can develop or flare up after some viral infections and some vaccinations (rubella, measles, and mumps) but is not observed with others [1,2]. The current SARS-CoV-2 (COVID-19) pandemic is characterized by severe pneumonia, multiorgan disease, and dysregulation of immunity [3,4]. De novo and exacerbation of autoimmune phenomena have been described both after COVID-19 infection [5,6,7,8,9] and vaccination [5,10,11,12,13,14], one of the most effective weapons to control disease [15]. Considering vaccination and ITP, so far, the relationship has not been well-characterized, and some data are still a matter of debate [16,17]. Such a state of uncertainty has raised concerns among both practitioners and patients, especially in the chronic ITP setting when determining when and how to proceed with anti-COVID-19 vaccination and how to set up a correct follow-up.

Furthermore, less is known about the serological response in ITP patients (pts) [18]. Not being an oncological disease nor an autoimmune disorder for which heavy immunosuppressive treatment is needed, so far, researchers have not focused on this specific population to study the passive immunization rate.

In order to evaluate the global impact of the COVID-19 pandemic on chronic ITP pts, we collected data on the platelet (plt) trend after COVID-19 infection and vaccination after the first, second, and third doses; furthermore, we analyzed the serological response after the first two doses of COVID-19 vaccination in chronic ITP patients regularly followed at our Institution.

## 2. Materials and Methods

The study retrospectively and prospectively evaluated all pts currently followed for chronic ITP [19] who experienced COVID-19 infection and/or received at least one dose of the anti-COVID-19 vaccine with an available plt count before and after these events.

The last patient was included in July 2022. Data regarding demographic features, ITP duration, and number and type of previous and current treatment were included. Mean plt count after infection/vaccination and reduction in plt count of any grade, described as absolute and percentage reduction, were recorded.

ITP exacerbations was defined as
(1)>50% drop in plt count compared with baseline;(2)>20% decline in plt count compared with baseline and a plt nadir < 30.000/uL;(3)Need for rescue medication.

Serological analysis for IgG anti-COVID-19 spike protein was carried out after second vaccine dose using LIAISON^®^ SARS-CoV-2 TrimericS IgG assay (DiaSorin S.p.A). Seroconversion cut-off considered was ≥33.8 BAU/mL (binding antibody units).

The study was approved by the local Ethical Committee (number 0000511 on 4 April 2022); included subjects signed informed consent.

The descriptive statistics for the demographic and clinical characteristics of patients were expressed as median and interquartile ranges (IQRs) for continuous variables and as absolute frequency (n) and percentage (%) for the categorical variables. Association between categorical variables was investigated using Pearson’s Chi-squared test (for cell frequency *n* ≥ 5) and Fisher’s exact test (for cell frequency *n* < 5). The crude odd ratio (OR) was used as association measurement. All statistical tests were 2-sided, with the significance level set at *p* ≤ 0.05. Analyses were performed using the R software environment for statistical computing and graphics (version 3.4.1; http://www.r-project.org accessed on 30 September 2022).

## 3. Results

Sixty-four chronic ITP pts regularly followed at our Institution were included in the analysis. The study population’s characteristics are summarized in Table 1.

Considering active COVID-19 infection, 11 pts were infected before any vaccination. Data on plt trends were available for six pts after a median time of 0 days (IQR 0-6) from infection detection. A reduction in plt count was observed in all six subjects (100%) with a median plt count after infection of 3.000/uL (IQR 2000–46500/uL), median absolute plt reduction of 116.500/uL (IQR 100.750–157.750/uL), and median percentage reduction of 96% (IQR 93.75–97.50%). ITP exacerbation was observed in five pts (83.3%); all cases required rescue therapy with Intravenous Immunoglobulin (IVIG) and steroids, and two pts were also treated with TPO receptor agonist (TPO-RA) introduction.

Sixty-one subjects received at least one dose of the anti-COVID-19 vaccine. An available plt count after the first dose was available for 41 pts after a median time of 10 days (IQR 5–13 days). A reduction in plt count was observed in 17 pts (41.4%) with a median plt count after the first dose of 110.000/uL (IQR 71.000–232.000/uL), a median absolute reduction of 34.000/uL (IQR 22.000–72.000/uL), and a median percentage reduction of 20% (IQR 15.2–48%). An ITP exacerbation after the first vaccine dose was observed in five pts (12.2%) with three pts (7.31%) requiring rescue treatment (one pt increased TPO-RA dose; one pt started TPO-RA, and the latter was treated with IVIG). In comparison to the plt trend after COVID-19 infection, a significant difference was observed in terms of plt reduction of any grade (OR 16.94, *p*-value = 0.009), ITP exacerbation (OR 36.0, *p*-value < 0.001), and rescue therapy necessity (OR = 63.33 *p*-value < 0.001).

Fifty-eight pts received a second vaccine dose; an available plt count after the second dose was available for 36 pts after a median time of 14 days (8–20 days). A reduction in plt count was observed in 16 pts (44.4%) with a median plt count after the first dose of 138.000/uL (IQR 78.500/uL–208.750/uL), a median absolute reduction of 20.000/uL (IQR 14.500–66.750/uL), and a median percentage reduction of 21% (IQR 6.75–40.25%). An ITP exacerbation after the second vaccine dose was observed in six pts (16.6%), with two pts (5.5%) requiring rescue treatment (TPO-RA dose increase). Only one of the exacerbated cases also experienced an exacerbation after the first vaccine dose. In comparison to the plt trend after COVID-19 infection, a significant difference was observed in terms of plt reduction of any grade (OR 15.00, *p*-value = 0.022), ITP exacerbation (OR 25.0, *p*-value = 0.003), and rescue therapy necessity (OR = 85 *p*-value < 0.001).

After the third vaccine dose, data regarding the plt trend were available for 33 pts after a median time of 16 days (IQR 10–46 days). A plt reduction of any grade was observed in 14 pts (42.4%) with a median plt count after vaccination of 171.000/uL (IQR 96.000–220.000/uL), a median absolute reduction of 30.500/uL (IQR 12.250–88.000/uL), and a median percentage reduction of 19% (IQR 4.96–43.78%). An ITP exacerbation was observed in only two subjects (6.06%), none of whom required rescue treatment. In comparison to the plt trend after COVID-19 infection, a significant difference was observed in terms of plt reduction of any grade (OR 16.29; *p*-value = 0.02), ITP exacerbation (OR 77.5; *p*-value < 0.001), and rescue therapy necessity (OR = 330.00; *p*-value < 0.001).

Twelve subjects experienced a COVID-19 infection after at least two vaccine doses. Considering 11 pts for which the plt count was available before and after infection, a plt reduction of any grade was observed in four pts (36.4%) after a median time of 8 days (IQR 2–14 days), with a median plt count after infection of 126.500/uL (IQR 86.250- 146.500/uL), a median absolute reduction of 23.000/uL (IQR 17.000–36.500/uL), and a median percentage reduction of 17.60% (IQR 15.30–36.20%). An ITP exacerbation with rescue therapy necessity (IVIG and steroids) was observed in two subjects (18.2%). In comparison to the plt trend after COVID-19 infection before any vaccination, a significant difference was observed in terms of plt reduction of any grade (OR 21.00; *p*-value = 0.035), ITP exacerbation (OR 22.50; *p*-value = 0.035), and rescue therapy necessity (OR = 22.50; *p*-value = 0.035).

A serological assay after the second dose was performed on 43 pts, after a median of 171 days (range 6–283 days) from the second vaccine dose. Median anti-Spike IgG titer resulted in a 293 BAU/mL (IQR 73.35–758.5) BAU mL). Seroconversion was detected in 36 pts (83.7%). Of the seven pts who did not achieve seroconversion after a median of 164 days from the second dose, all were currently on treatment for ITP (five on TPO-RA, one on TPO-RA and steroids, and one on AZT); three were over 75 years of age; three presented an associated immune cytopenia (two Evans syndrome and one cyclic neutropenia), and only one pt had been previously splenectomized. None of them were infected before vaccination. One 79-year-old subject with no seroconversion after the two vaccine doses became infected with COVID-19 and was treated with Molnupiravir. The patient did not develop any COVID-19-related symptoms or complications and did not experience an ITP exacerbation.

## 4. Discussion

Our report is the first dynamic description of the impact of the COVID-19 pandemic in chronic ITP pts focusing on three different scenarios: COVID-19 infection before any vaccination, after COVID-19 vaccination (considering the first, second, and third doses), and COVID-19 infection after vaccination. We also analyzed the frequency of seroconversion in this population.

Infections are one of the well-known precipitating factors for relapses and exacerbations in pts with ITP. Very few studies focusing on the effect of COVID-19 infection in chronic ITP patients have been published [5,6,7,8,9,20,21] with divergent conclusions and an unclear distinction between de novo ITP and previous/chronic ITP. One of the largest studies [7] reported the outcome of 32 pts with a pre-existing ITP with COVID-19 infection reporting a reduction in plt count in 47% of the pts, all requiring treatment or a modification of current ITP treatment. Our data confirm the high frequency of ITP exacerbation after COVID-19 infection in nonvaccinated subjects, as all pts for which data on the plt trend were available experienced a plt reduction of any grade and an ITP exacerbation requiring rescue treatment in 83% of the infected.

Considering the plt trend after COVID-19 vaccination, different monocentric and multicenter experiences have been published. Considering the largest studies available focusing on pre-existing ITP and applying the same definition of exacerbation used in our analysis [11,12,22], a plt reduction of any grade was reported in 30–55% of the cases and an ITP exacerbation in 6.0% to 20.0% of the cases. Possible risk factors for ITP exacerbation identified were plt baseline, recent ITP treatment initiation, younger age [12], previous splenectomy, and more than five prior therapy lines [11]. Our data substantially confirm those published as a plt reduction of any grade was observed in 41.0 to 44.0% of the study population, but an ITP exacerbation was observed in only 12.0 to 20.0%, and pts who required rescue treatment amounted to only 5.0 to 7.0%. We also collected data on the plt trend after the third dose, and we observed a lower incidence in plt reduction and ITP exacerbation compared to the first and second doses, with no pts needing rescue treatment.

The most interesting aspect of our analysis is the comparison in terms of the plt trend in these different situations: as reported, a significant difference in terms of ITP exacerbation can be observed between infected pts before vaccination (83.0%) and vaccinated ones (13.8–14.6%) as well as for rescue treatment necessity (83.0 vs. 7.0%). A difference can also be observed for infected pts before and after vaccination: subjects infected after vaccination experienced a lower rate of ITP exacerbation and rescue treatment necessity (83.0% vs. 18.2%) (Figure 1, Figure 2 and Figure 3).

Besides supporting vaccination to reduce the risk of severe COVID-19 disease, these results suggest a further benefit of the COVID-19 vaccines for ITP pts. Indeed, a reduction in plt count can be observed after both COVID-19 infection and vaccination, but a higher risk of ITP exacerbation and rescue treatment necessity was observed after COVID-19 infection in nonvaccinated pts, while after COVID-19 vaccination, the incidence of severe ITP exacerbation remains low. Furthermore, the risk of ITP exacerbation in the case of subsequent infection is also reduced.

The immunogenicity of COVID-19 vaccines on immunocompromised pts is mainly available for oncologic pts or those affected by rheumatological disease [23,24,25]. Moreover, immunocompromised pts have mainly been excluded from vaccine anti-COVID-19 clinical trials. Autoimmune cytopenia pts are a different category not directly comparable to oncological or severe immunocompromised pts, even if some therapies are used in both groups (steroids, Rituximab, Azathioprine, etc.) Seroconversion data on immune cytopenia pts are lacking and not easily comparable, considering the different laboratory assays used or the different timing in antibody measurements. Fattizo et al. [18] prospectively reported on the outcome in terms of antibody development of pts suffering from different immune cytopenia and bone marrow failure syndromes, among whom are 25 ITP pts. The authors observed that 96% of ITP pts presented a positive antibody response after 2+/−1 months after the second dose. In our analysis, an inferior rate of seroconversion was observed (83%, 88% if excluding Evans syndrome pts). However, in contrast with the previous study, our analysis was conducted after a median time of 5 months from the second dose. Thus, such results could suggest either a seroconversion failure or an early loss of response. Of the seven non-seroconverted pts, three pts were older than 75, confirming the reduced immune response in older patients. Three different pts have an associated immune cytopenia other than ITP (2 Evans syndrome and one cyclic neutropenia) suggesting that an underlying complex autoimmune substrate could alter vaccine response, as observed by Fattizzo et al.

Our study carries some limitations. The observational nature and the pandemic emergency (with limited outpatient visits and difficulties to access the blood count check in case of infection) did not allow us to establish definite time points in plt count control after COVID-19 infection, vaccination, or serological assay. The monocentric nature limited study numerosity and did not allow the identification of possible risk factors for ITP exacerbation after infection or vaccination, nor for serological response failure. The majority of pts considered in the analysis received the Comirnaty, Pfizer Biontech vaccine, so our conclusion should be limited to only a portion of the vaccinated ITP pts; no comparisons were allowed among different vaccine types.

## 5. Conclusions

At the time of study submission, this was the first study prospectively analyzing the global impact of the COVID-19 pandemic, specifically focusing on chronic ITP pts and comparing COVID-19 infection with COVID-19 vaccination. Beyond the indubitable protective effect against severe infection, our data encourage COVID-19 vaccination in chronic ITP pts. Thus, a relatively low incidence of ITP exacerbation following vaccine sessions was observed in comparison with a high frequency of exacerbation and rescue treatment necessity after COVID-19 infection in unvaccinated pts. Moreover, the lower rate of ITP exacerbation cases observed in infected pts previously vaccinated suggests further protective effects in this population. Finally, a high seroconversion rate was observed, confirming data reported in previously published studies on immune cytopenia and rheumatological diseases, but more evidence is awaited to establish the clinical impact of the serological response.

## Figures and Tables

**Figure 1 biomedicines-10-02674-f001:**
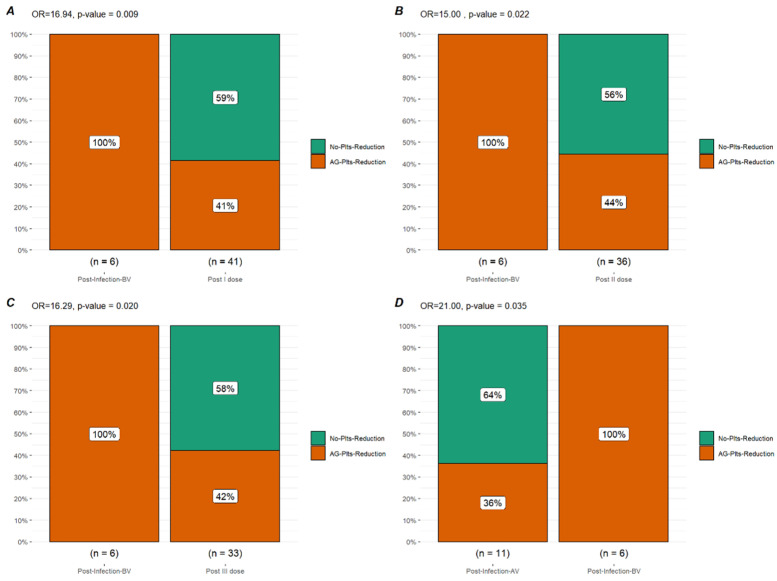
**Bar plot of platelet reduction of any grade**. Comparison in terms of platelet reduction of any grade (AG-Plt-Reduction) between infection before vaccination (Post-Infection-BV), Post first vaccine dose (Post I dose (**A**)), second dose (Post II dose) (**B**), and third dose (Post III dose) (**C**) and post-infection after vaccination (Post-Infection-AV) (**D**).

**Figure 2 biomedicines-10-02674-f002:**
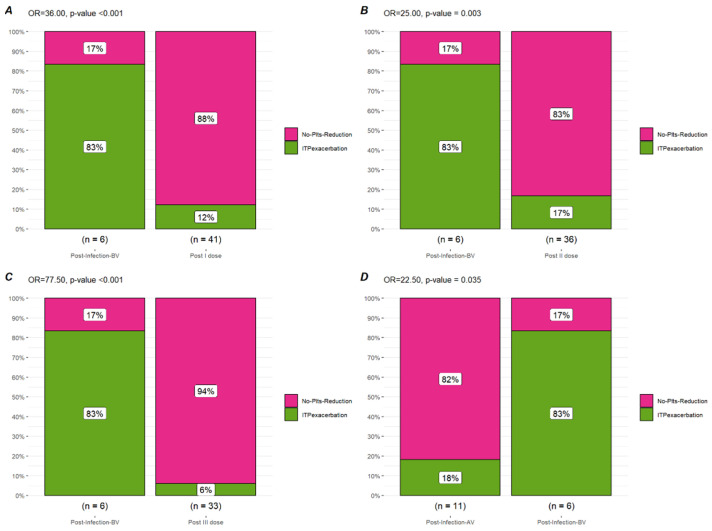
**Bar plot of ITP exacerbation**. Comaprison in terms of ITP exacerbation between infection before vaccination (Post-Infection-BV), Post first vaccine dose (Post I dose) (**A**), second dose (Post II dose) (**B**), and third dose (Post III dose) (**C**) and post-infection after vaccination (Post-Infection-AV) (**D**).

**Figure 3 biomedicines-10-02674-f003:**
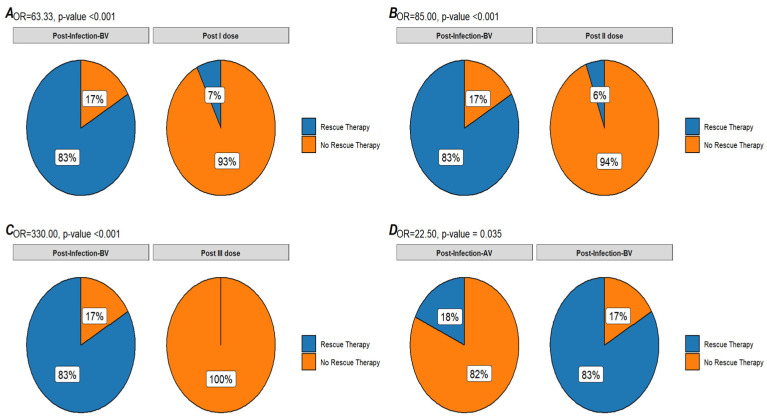
**Round plot of rescue therapy necessity**. Comparison in terms of rescue therapy necessity between infection before vaccination (Post-Infection-BV), Post first vaccine dose (Post I dose) (**A**), second dose (Post II dose) (**B**), and third dose (Post III dose) (**C**) and post-infection after vaccination (Post-Infection-AV) (**D**).

**Table 1 biomedicines-10-02674-t001:** **Study Population Features**. Descriptive statistics of sample features. ITP—Immune Thrombocytopenic Purpura. TPO-RA—TPO Receptor Agonist. SD—standard deviation.

Study Population Features	N = 64
Age (years), mean (SD)	62.39 (17.26)
Gender, n (%)	
Male	23 (35.90)
Female	41 (64.10)
Number of previous ITP treatments, median	2
Previous splenectomy (%)	15 (23.40)
On therapy at infection/vaccination time, n (%)	44 (68.75)
Steroids	1 (1.56)
Steroids and TPO-RA	3 (4.68)
TPO-RA	37 (57.81)
Azathioprine	2 (3.12)
Azathioprine and TPO-RA	1 (1.56)
Off Therapy at infection/vaccination time, n (%)	20 (31.25)
First Dose Anti-SARS-CoV-2 Vaccination, n (%)	61
Comirnaty, Pfizer Biontech	52(85.24)
Spikevax, Moderna	8 (13.10)
Vaxzevria, Astrazeneca	1 (1.64)
Second Dose Anti-SARS-CoV-2 Vaccination, n (%)	58
Comirnaty, Pfizer Biontech	50 (86.20)
Spikevax, Moderna	8 (13.79)
Third Dose Anti-SARS-CoV-2 Vaccination, n (%)	34
Comirnaty, Pfizer Biontech	27 (79.41)
Spikevax, Moderna	7 (20.58)

## Data Availability

Not applicable.

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
