# Peer review of "Impact of COVID-19 Infection, Vaccination, and Serological Response in Immune Thrombocytopenic Purpura Patients: A Single-Center Global Analysis"

_biomedicines, 2022, doi:10.3390/biomedicines10112674_

Round 1
Reviewer 1 Report
The paper is interesting and well written. I suggest to discuss the impact of vitamin D and microbioma on immune responses including autoimmunity, the role of IL-31/IL-33 axis and Th17 cells in autoimmune responses
Author Response
Kind Reviewer
Thank you for you comment. The impact of Vitamin D, microbioma, IL-31/IL-33 axis and the role of Th17 in autoimmune responses are very interesting aspects. However, our study is an observational one and such insights are not included in standard clinical practice ; for these reasons they cannot be add to our analysis. They would be an intersting aspects to elaborate in future interventional studies for sure.
Tank you again for the comment
Reviewer 2 Report
Thank you for giving me the opportunity to read and comment a report “Impact of COVID-19 infection, vaccination and serological response in immune thrombocytopenic purpura patients: a single Center global analysis”, by Dainese C., et al.
In the reviewed manuscript, the global impact of the COVID-19 pandemic on chronic immune thrombocytopenic purpura patients has been evaluated.
This paper is well written, correctly structured with a suitable research concept, the study limitations are addressed, and it is of relevance to readers of the journal.
However, I include a few comments for your consideration.
· The introduction section is very short, just a single paragraph, obviating the objectives. In the opinion of this reviewer, it would be necessary to describe in more depth the current state of the problem.
· It would be appropriate to standardize the acronym COVID-19, since it appears in different forms: COVID-19, COVID19 and COVD19.
· The acronym “Sars” should be capitalized (line 39).
· According to the rules of the journal, there must be a space between the word and the bibliographic reference.
· There are not any sub-sections under Methods section, which also makes it difficult to understand the study methods.
· The use of the mean and median in the results is very confusing. Sometimes the authors use the mean with the range (line 78), sometimes the mean with the IQR (line 80), sometimes the median with the range (line 130), and finally the median with the IQR (line 131). As indicated in the material and methods section, the mean is accompanied by the SD and the median by the IQR. Therefore, it would be advisable for the authors to revise the results.
Author Response
Kind reviewer,
Thank you very much for your helpful considerations. We have proceed with the correction you suggested, in details:
- We implemented the introduction section, but we think the depth of the problem is more specified in the discussion paragraph.
- We standardized the acronym COVID-19.
- We capitalized the acronym SARS in line 39.
- We added a space between words and bibliographic references.
- The journal did not specified subsections in the methods paragraph, however we tried to make it more schematic and reader friendly
- We corrected the use of mean and median. We reported median value with IQR.
Thank you again for your attention.